# Adsorption Mechanism of High-Concentration Ammonium by Chinese Natural Zeolite with Experimental Optimization and Theoretical Computation

**Pan Liu** [1,2], **Aining Zhang** [1], **Yongjun Liu** [1,*], **Zhe Liu** [1], **Xingshe Liu** [1], **Lu Yang** [1] **and Zhuangzhuang Yang** [1]

[1] School of Environmental and Municipal Engineering, Xi'an University of Architecture and Technology, Xi'an 710055, China
[2] Xi'an Aeronautical Polytechnic Institute, Xi'an 710089, China
[*] Correspondence: liuyongjun@xauat.edu.cn

**Abstract:** Natural zeolite, as an abundant aluminosilicate mineral with a hierarchically porous structure, has a strong affinity to ammonium in solutions. Adsorption mechanism of high-concentration ammonium (1000~4000 mg-N/L) in an aqueous solution without pH adjustment onto Chinese natural zeolite with the dosage of 5 g/L was revealed by the strategy of experimental optimization integrated with Molecular Dynamics (MD) simulation, and found the maximum ammonium adsorption capacity was 26.94 mg/g. The adsorption kinetics and isotherm analysis showed that this adsorption process fitted better with descriptions of the pseudo-second-order kinetics and Freundlich model. The theoretical calculations and infrared-spectrum characterization results verified the existence of hydrogen bonds and chemisorption. Therefore, the adsorption mechanism by natural zeolites of high-concentration $NH_4^+$ is defined as a process under the joint influence of multiple effects, which is mainly promoted by the synergy of the ion exchange process, electrostatic attraction, and chemisorption. Meanwhile, the hydrogen bond also plays an auxiliary role in this efficient adsorption. This study presents important theoretical significance for enriching the mechanism of zeolites adsorbing $NH_4^+$ from water, and provides reference and theoretical guidance for further exploring the potential application of natural zeolites.

**Keywords:** adsorption mechanism; ammonium; natural zeolite; ion-exchange; molecular dynamics simulation; physisorption; chemisorption




## 1. Introduction

Natural zeolites are crystalline hydrated aluminosilicate minerals with numerous pores filled with alkali, alkaline earth cations, and water, and possess valuable physico-chemical properties such as unique selective sorption, ion exchange, and thermal stability. Natural zeolites have been broadly applied for wastewater treatment [1–7].

Previous studies have mainly focused on the applications of natural, modified, and synthetic zeolites for the removals of low-concentration (<200 mg-N/L) and trace ammonium within municipal sewage, groundwater, or tap water [8–11]. Generally, the air stripping method is adopted for high-concentration ammonium removal [12,13]. However, air stripping usually requires elevated alkalinity and temperature to increase the proportion of gaseous $NH_3$, this demands continuous addition of acids to reduce the alkalinity before discharge, which pollutes the atmosphere severely and increases the operating cost [14]. Compared to other techniques, the adsorption has numerous favorable characteristics, such as a high affinity towards ammonium, low-cost, convenience for engineering configurations as well as environmental friendliness [15]. In addition, natural zeolites have been abundant in nature. These advantages make it quite functional to remove high-concentration ammonium from industry wastewater both effectively and environmentally.

There have been frequent reports that when zeolites adsorbed low-concentration $NH_4^+$, processes of ions exchange and adsorption are the main pathways [16–18]. Among them, the ion exchange process is defined as an outside surface complexation, including the replacement of cations in zeolite structure. The adsorption process is considered an inside surface complexation. The adsorbed cations form a bond with the active functional group of zeolites or are drawn by electrostatic attraction [16]. The research on the adsorption of high-concentration $NH_4^+$ has been involved in some works of literature also, but it was usually employed as the control group for other research on the adsorptive performance of low-concentration $NH_4^+$ without being specifically studied as the focus, while the reported research on its adsorption mechanism was even less. In 2013, L. Lin et al. found that ion exchange was the dominant mechanism of ammonium adsorption, but the order of exchange selectivity was distinct in ammonium solutions of different concentrations ranges (10~4000 mg-N/L) with $Na^+$ as the predominant ions for ammonium at low levels while $Ca^{2+}$ exceeding $Na^+$ at high levels (>1000 mg-N/L) [19]. This indicates that zeolites exhibit different adsorption behaviors at different concentration ranges of ammonium, however, the adsorption mechanism at high concentrations remained not explored further within this study. M. J. Manto et al. investigated the ammonium recovery in 1000 mg-N/L aqueous solution with ZSM-5 as the sorbent, and observed that its uptake of ammonium was much higher than its Cation Exchangeable Capacity (CEC) on the ZSM-5 [20]. The formation of hydrogen bonding between $NH_4^+$ and the framework of oxygen might cause the excess of CEC [21], which lay the foundation for the synergistic adsorption mechanism of zeolite in the adsorption of high-concentration $NH_4^+$ with multiple forms of interactions. Both groups of Ye L. et al. and O'Connor et al. confirmed that the multiple hydrogen bonding between the guest ammonium and oxygen atoms were formed in Molecular Dynamics (MD) simulation's process of zeolites capturing ammonium [21,22]. These results imply the maximum possibility of the multiple interactions coexistence including physisorption and chemisorption, and have great essentiality an for in-depth investigation of the complex adsorption mechanism of high-concentration ammonium by zeolites.

In this paper, the natural zeolites' adsorption mechanism of high-concentration $NH_4^+$ as the research objective, was investigated systematically with the research method integrating experiments with theoretical calculations, rely on adsorption kinetics/isotherms/thermodynamics and molecular dynamics simulation of the high-concentration $NH_4^+$ adsorption on the natural Chinese zeolite. This study not only attaches important theoretical significance to enriching the research on zeolites' adsorption mechanism of high-concentration $NH_4^+$ from water, but also provides a reference and theoretical guidance for further exploring the potential application of natural zeolites.

## 2. Materials and Methods

### 2.1. Materials

The Chinese natural zeolite from Zhengzhou in Henan Province was ground into particles with the size ranging from 0.5–1.0 mm. Before being used for batch experiments, these zeolite particles were rinsed with deionized water and dried in an oven at 105 °C overnight. Analytical reagent grade of ammonium chloride and sodium chloride were both purchased from Tianjin Chemical Reagent Co., Ltd. from Tianjin City in China.

### 2.2. Characterization Techniques

The morphology of natural zeolite was characterized by Scanning-Electron Microscope (SEM, ZEISS, MERLIN Compact, Germany), its chemical composition was characterized by X-ray Energy Dispersion Spectrometry (EDS, Bruker 6/30, Bruker Corporation, USA), and its mineral species were characterized by an X-Ray Diffractometer (XRD, Ultimate IV, Rigaku Corporation, Japan). The specific surface area, pore volume, and pore size distribution of natural zeolite were measured through the $N_2$ adsorption-desorption experiment with a heating rate of 10 °C/min at 77 K via an Automatic Volumetric Sorption Analyzer (ASAP 2460, Micromeritics instrument Ltd. America). Furthermore, the specific surface

areas were computed by employing the Brunauer–Emmett–Teller (BET) method. The total pore volume was estimated in terms of the amount of adsorption at a relative pressure (P/P$_0$) of 0.95 by using the Barrette–Joynere–Halenda (BJH) method. Pore size distributions were also measured using the BJH method.

### 2.3. Liquid-Phase Adsorption Experiments

Firstly, 5 g of Chinese natural zeolite and 100 mL of NH$_4$Cl solution at a prescribed concentration (100~4000 mg-N/L) without pH adjustment were placed in a 250 mL Erlenmeyer flask, and stirred at a specified temperature in the thermostatic shaker (HZQ-X3000, Shanghai Yiheng Scientific Instrument Co., Ltd., Shanghai, China) for several hours. Then the suspension after adsorption was filtered via a 0.22-μm filter and the filtrate was analyzed for the residual ammonium's concentration by Nessler's reagent spectrophotometry method at 420 nm with a UV/Visible spectrophotometer (UV2150, UNICO (Shanghai) Instrument Co., Ltd., Shanghai, China).

### 2.4. Simulation of Zeolites' Ammonium Capture and Bonding Strength

In this study, the interaction between a rigid ammonium ion and zeolite framework was the focused objective. All the structures (consisting of zeolite substrates, ammonium ions and water molecules) were determined by Density Functional Theory (DFT) method using the Dmol$^3$ program package in Materials Studio 2018, and the initial zeolite structure shown in Figure S1a was adopted to model the geometry of the framework. The exchange and correlation effects were both described by Perdew-Burke-Ernzerhof (PBE) functionals [23], through a Generalized Gradient Approximation (GGA) approach. Subsequently, the Hirshfeld charge analysis of the optimized structures was carried out, and these stably charged structures were applied to the dynamic simulation via the Forcite module.

The high-concentration ammonium's solution model was built with the Amorphous Cell module, and the initial cubic simulation lattice of the system was set as: x = 39.17 Å, y = 39.96 Å, and z = 240 Å. There was a 2-nm-thick vacuum layer on top of the solution to weaken the effect of the zeolite substrate within the upper periodic lattice, as shown in Figure S1b. The distribution of the random positions of ammonium on the pore surface was depicted in Figure S1c. The Simple Point Charge (SPC) model [24], for describing the aqueous solution environment accurately [25], was employed on all water molecules. All Molecular Dynamics simulations were performed in the COMPASS force field [26,27] and equilibrated at constant temperature (308.15 K) with a volume (NVT) of 10 ns.

## 3. Results

### 3.1. Characterization of the Chinese Natural Zeolite

The morphologies of the Chinese natural zeolite were observed by SEM. As shown in Figure 1a, the zeolite presented a lamellar structure with macropores at the intersections among the slices of different orientations, which was characterized by geometrically well-defined channels and cavities, this largely governed the mobility and siting of interstitial ions and their availability for exchanges. The natural zeolites were examined by XRD to verify their mineral species and their contents were identified by the Rietveld method. As revealed in Figure 1b, the chosen zeolite is rich in heulandite (wt%: 57.43%) besides sanidine, tridymite and quartz.

The typical Class-IV isotherm with a fairly clear plateau was observed from the N$_2$ adsorption-desorption isotherm by the BJH pore size analysis (Figure 1c). The desorption curve showed an extensive range of mesopore structures attributed to plate-like particles with slit-shaped pores [28]. The conclusion was supported by the pore size distribution shown in Figure 1d where substantial mesopores dominated the zeolites' skeleton structure and some micropores together with a few macropores were also presented. As illustrated in Table 1, the mesopore volume accounted for a major part of the total pore volume, and the total pore volume of the zeolite was not large enough either (V$_p$ = 0.0586 cm$^3$/g). However, its adsorption capacity was much stronger than that of some reported zeolites (Table 1).

A higher percentage of mesopore volume might more favorable for $NH_4^+$ adsorption on zeolite than micropores. The low Si/Al ratio of 4.51 in the framework of the chosen Chinese natural zeolite determined by EDS (Table 2) also contributed to its high adsorption capacity because of its great affinities with ammonium ions. Besides, the natural zeolite was rich in K metal cations, as illustrated in Table 2.

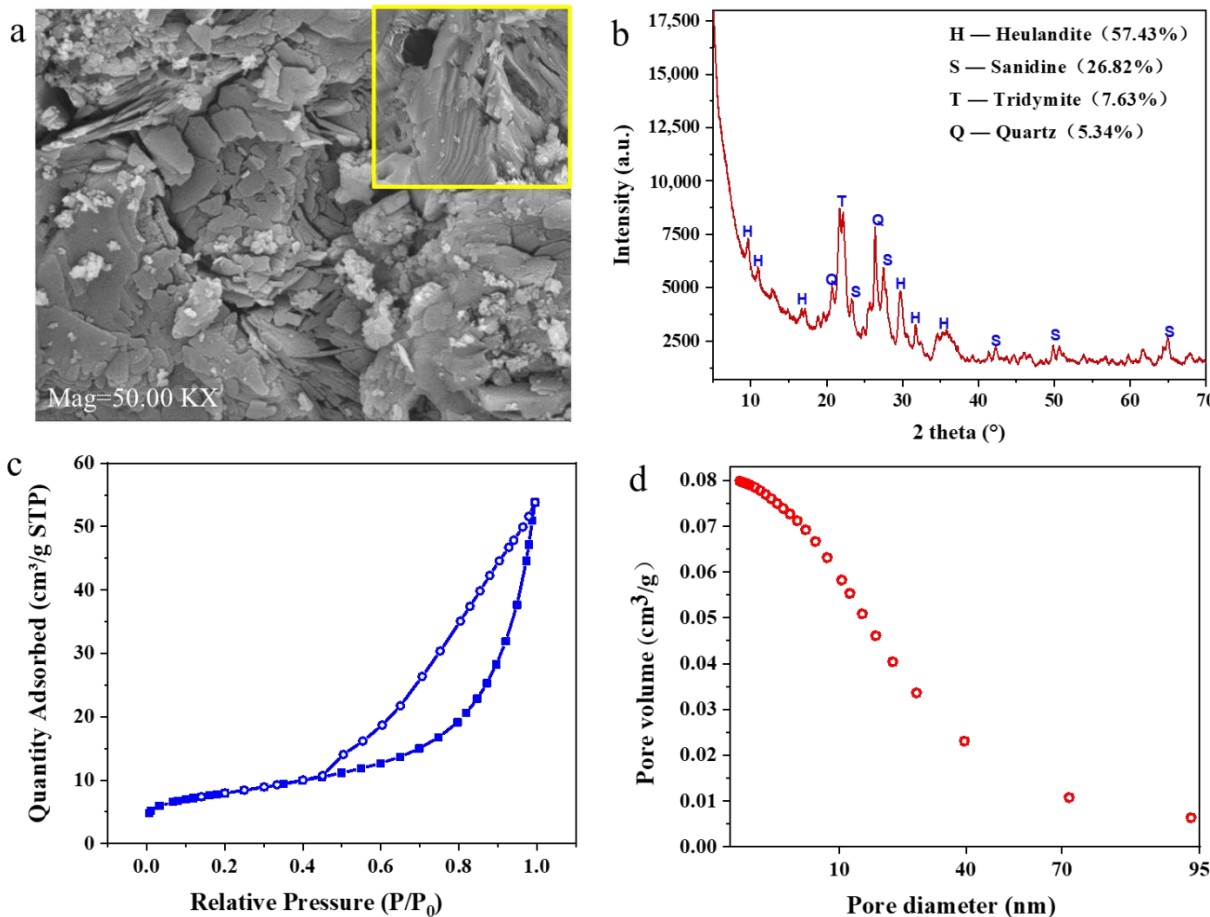

**Figure 1.** The basic properties of the natural zeolite. (**a**) SEM images; (**b**) XRD patterns; (**c**) N$_2$ adsorption-desorption isotherms at 77 K; (**d**) BJH pore size distribution.

**Table 1.** Physical properties of the Chinese natural zeolite in comparison with other zeolites.

| Sample Name | Capacity (mg/g) | BET Surface Area (m²/g) | Mean Pore Diameter (nm) | Micropores Volume (cm³/g) | Mesopores Volume (cm³/g) | Total Volume (cm³/g) | Reference |
|---|---|---|---|---|---|---|---|
| Chinese natural zeolite | 26.94 | 28.15 | 11.87 | 0.00271 | 0.0559 | 0.0586 | this study |
| Natural zeolite | 14.3 | 14.33 | 16.19 | | | 0.044 | [19] |
| NaCl-modified zeolite | 17.3 | 60.83 | 29.24 | | | 0.065 | [19] |

**Table 2.** Chemical compositions of the Chinese natural zeolites by EDS.

| Chemical Elements | wt.% | Chemical Elements | wt.% |
|---|---|---|---|
| O | 47.12 | Na | 0.82 |
| Si | 26.87 | Ca | 0.77 |
| Al | 5.72 | Mg | 0.34 |
| K | 2.76 | others | 14.2 |
| Fe | 1.40 | | |

### 3.2. Adsorption Characteristics of the Chinese Natural Zeolite

The effects of various influencing factors on ammonium removal processes were assessed systematically (Figure 2a,b) when a different dosage of Chinese natural zeolite was added into 100 mL of 4000 mg-N/L NH$_4$Cl solution without pH adjustment at different temperatures. The adsorption capacity and rate of ammonium increased with the enhancement of the zeolite dosage and temperature in this study. This enhancement might be related to the predicted increase in the zeolite receptor sites, ion-exchange sites, and the total surface area with the incorporation of higher masses of it in the addressed system [29]. On the other hand, the temperature had an important effect on ammonium removal since it could impact the diffusion characteristics of ammonium ion, because the adsorbate ammonium has stronger diffusion power and the adsorption site could be more active at higher temperatures [30]. When comprehensively considering the energy-consumption, environmental-friendly requirements and other factors, the optimal experiment was conducted with a zeolite dosage of 5 g/100 mL at 35 °C for different initial concentrations of ammonium (100~4000 mg-N/L) in an aqueous medium without pH adjustment.

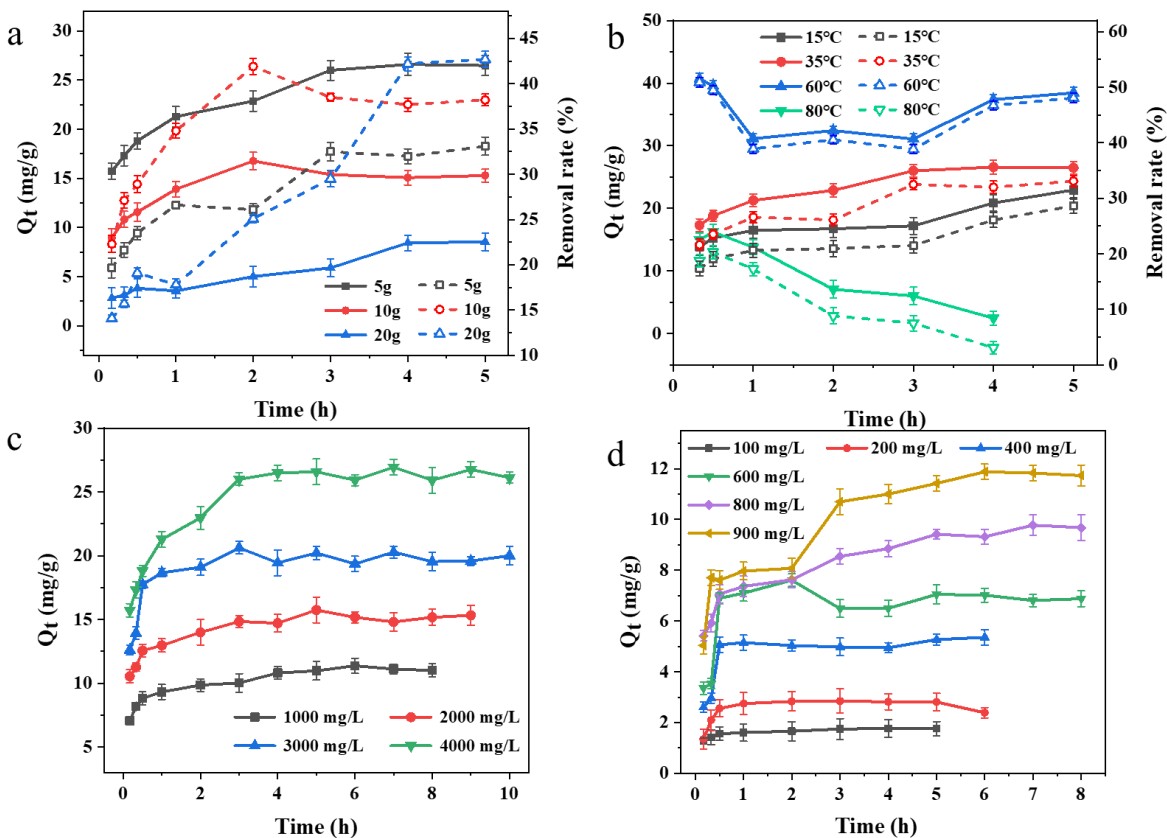

**Figure 2.** The effect of various treatment processes on the high-concentration ammonium removal. (**a**) Natural zeolite dosage; (**b**) Temperature; (**c**) High-concentration ammonium (1000~4000 mg-N/L) (**d**) Low-concentration ammonium (<1000 mg-N/L). The solid lines and dotted lines represent the adsorption capacities of the natural zeolite and the ammonium removal percentages, respectively in Figure 1a,b.

To explore the effect of high-concentration ammonium on the adsorption behavior of natural zeolites, NH$_4^+$ initial solutions with different concentration ranges were selected in this study. The dosage of zeolites was set to 5 g/100 mL, and the adsorption experiments of these natural zeolite were carried out at 35 °C. As illustrated in Figure 2c,d, the adsorption characteristics of natural zeolites to high-concentration ammonium were distinct from those of low-concentration ammonium. When the concentration exceeded 1000 mg/L, the natural zeolites could barely maintain a fixed adsorption capacity, indicating the inequilibrium

between the adsorption and desorption processes of $NH_4^+$, due to the desorption rate being larger at certain moments. The reason for this phenomenon might be the unique lamellar structure of the natural zeolite enabling $NH_4^+$ ions to easily diffuse within the pores. Nevertheless, the net adsorption capacity of zeolite to $NH_4^+$ was increasing rapidly. When it was adsorbed for 3 h, its adsorption capacity basically achieved the maximum value of 26.94 mg/g. Based on the related literature, ammonium adsorption capacity by natural zeolite was generally 2.7~30.6 mg/g [31]. Besides, by comparing this adsorption capacity to those of other zeolites (Table 3), it was found that our Chinese natural zeolite presented an excellent adsorption capacity to ammonium. Certainly, this differential adsorption behaviors of natural zeolite to ultrahigh concentration $NH_4^+$ were ascribed to its different ways of action when adsorbing low and high concentrations of $NH_4^+$. The possible reason for this unstable adsorption capacity and significant desorption phenomenon was the presence of interactions between $NH_4^+$ and the zeolite surfaces other than the commonly-regarded ion-exchange process and electrostatic attraction, which destabilized the $NH_4^+$ existence on the zeolite surfaces.

**Table 3.** Adsorption characteristics to ammonium by the selected Chinese natural zeolite and other reported materials.

| Samples | $C_0$ (mg/L) | Equilibrium Time (min) | $Q_{exp,max}$ [1] (mg/g) | $Q_{Lan,max}$ [2] (mg/g) | Reference |
|---|---|---|---|---|---|
| Natural zeolite | 1000~4000 | 180 | 26.94 | 27.06 | this study |
| Iranian zeolite | 90~3620 | 60 | 13.27 | 11.52 | [32] |
| Natural Chinese (Chende) zeolite | 50~300 | 180 | 9.41 | 9.41 | [33] |
| Na-Yemeni natural zeolite | 10~250 | 20 | 11.2 | | [34] |
| Modified bentonite | 0~350 | 60 | 5.85 | 5.8503 ± 0.08 | [18] |

[1] $Q_{exp,max}$ denote the experimental maximum adsorption capacity. [2] $Q_{Lan,max}$ denote the Langmuir maximum adsorption capacity.

### 3.3. Total Ion Exchange Capacity and Selectivity for Ammonium

The ammonium's ion-exchange selectivity for metal cations in the framework was investigated in the 4000 mg-N/L solution. Assuming that the exchangeable cations in zeolites were $Na^+$, $K^+$, $Ca^{2+}$ and $Mg^{2+}$ [35], the contents of exchanged different metal cations at different adsorption time was monitored by atomic absorption spectrometry. As seen in Figure 3a, the ammonium's ion-exchange selectivity for metal cations onto the natural zeolite evidently followed the order of $Ca^{2+} > Na^+ > K^+ > Mg^{2+}$ at first 3 h, which was distinguished from the other order of $Na^+ > K^+ > Ca^{2+} > Mg^{2+}$ observed by Watanabe et al. [36]. This inconsistency was attributed to the difference in chemical compositions of their zeolites containing a rather low content of Ca and the initial low-concentration of ammonium ($\leq$500 mg-N/L). However, Lin et al. have also found that $Ca^{2+}$ started to dominate the ion exchange process when the initial ammonium concentration was increased to above 1000 mg-N/L [19], indicating that the ion exchange selectivity of high-concentration ammonium was completely distinguished from that of low-concentration ammonium's solution. Subsequently, the released $Na^+$ and $K^+$ increased greatly after exchangeable $Ca^{2+}$ ions were exhausted, and the order changed to $Na^+ > K^+ > Ca^{2+} > Mg^{2+}$. The interpretation of a slight decrease in soluble $Ca^{2+}$ content was the released $Ca^{2+}$ being re-exchanged to zeolites with $Na^+$ and $K^+$ because of its strong affinity with the zeolite framework. The low content in zeolite and highly hydrated radius in the solution inhibited the exchanges of $Fe^{2+}$ and $Mg^{2+}$ with ammonium ions [37].

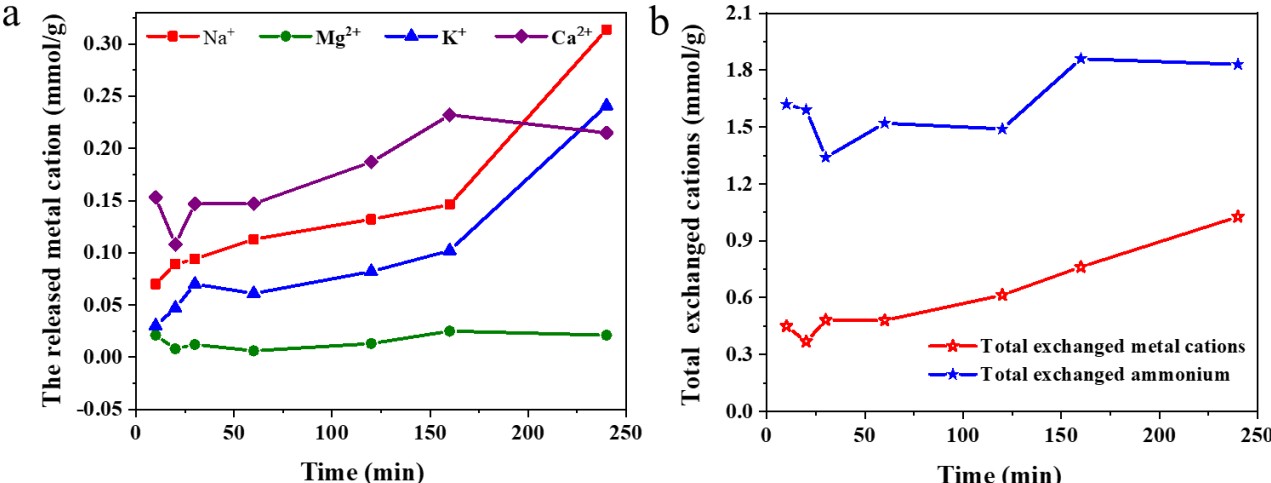

**Figure 3.** The exchange of the metal cations in natural zeolite with ammonium in solution. (**a**) Released metal cations contents; (**b**) Comparison of total exchanged ammonium and metal cations contents.

If only the strong electrostatic attraction was effective for realizing the high capacity of ammonium by the low ratio of Si/Al in the natural zeolite skeleton, the total capacity of loading ammonium on zeolites should equate to the sum of exchanged cations. This sum is defined as the Ion Exchange Capacity (IEC) and is expressed as the following formula:

$$IEC = [Na^+] + [K^+] + 2[Ca^{2+}] + 2[Mg^{2+}] = [NH_4^+]$$

The calculated IEC was compared with the total of reduced ammonium concentration and was plotted in Figure 3b which showed that the ammonium adsorption capacity was much higher than the actual IEC at any time interval, demonstrating that other interacting forces coexisted with electrostatic attraction and exerted a greater effect on efficient ammonium removal. The existence of other interactions would be supported by the conclusions of adsorption kinetics, equilibrium, and MD simulation.

*3.4. Adsorption Kinetics and Isotherm Properties*

Pseudo-first-order, Pseudo-second-order, and Elovich models were chosen to investigate the kinetic properties of the natural-zeolite-ammonium adsorption system, and their expressions were individually listed in Table S1. As seen in Figure 4a and Table 4, the adsorption to ammonium followed the pseudo-second-order model. The pseudo-second-order model simulated reactions with more chemical affinity between ammonium ions and zeolite surfaces [38,39]. Besides, according to the fitting results, it could also be found that when natural zeolite adsorbed ultrahigh-concentration $NH_4^+$, its kinetic fitting deviation turned larges. This was ascribed to the complex adsorption behavior of high concentration pollutants and the synergy of multiple adsorption effects, leading to a larger deviation when fitting to the theoretical model with fewer parameters above. Generally, when the initial concentration was 4000 mg/L, the adsorption process suited better with the description by the pseudo-second-order kinetics model, indicating that the natural zeolites' adsorption process to ultrahigh concentration $NH_4^+$ was more consistent with the chemisorption mechanism.

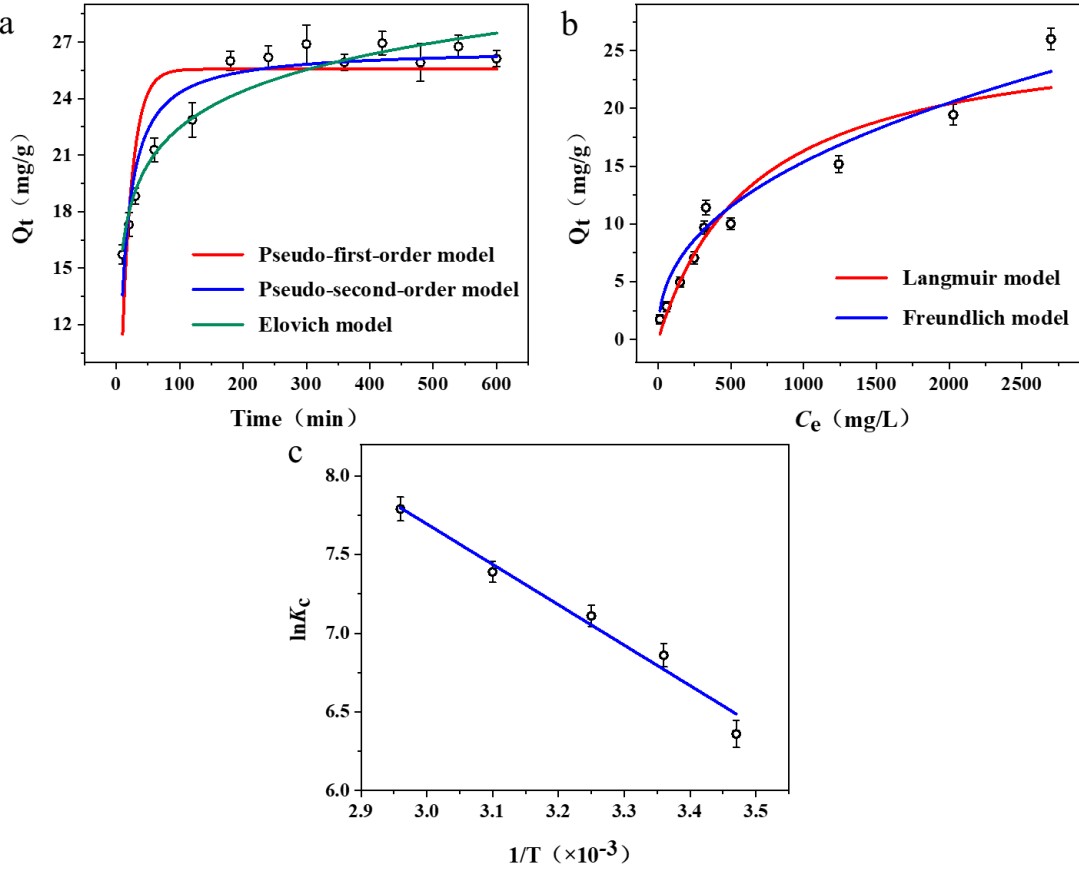

**Figure 4.** (**a**) Kinetic model fitting; (**b**) Isotherm Fitting; (**c**) Thermodynamic Fitting.

**Table 4.** Different parameters and errors values associated with isotherm models.

| | Langmuir | | | Freundlich | | |
|---|---|---|---|---|---|---|
| $q_m$ | $K_L$ | $R^2$ | | $K_F$ | $1/n$ | $R^2$ |
| 27.06 | 0.0015 | 0.935 | | 0.872 | 0.415 | 0.969 |
| | Pseudo First-Order | | | Pseudo Second-Order | | |
| $q_e$ | $k_1$ | $R^2$ | | $q_e$ | $k_2$ | $R^2$ |
| 25.56 | 0.059 | 0.754 | | 26.65 | 0.016 | 0.927 |

Next, the adsorption equilibria properties of natural zeolite to ammonium ions were assessed by the results of the nonlinear fittings with the Langmuir model and Freundlich model (Figure 4b). The illustrative equations of the inspected models were documented in Table S1 and the isotherm parameters were listed in Table 4, respectively. The Langmuir model hypothesized the homogeneous and monolayer adsorptions to ammonium by natural zeolite, while the Freundlich model presumed the heterogeneous and multi-layer uptakes [40,41]. Considering the values of $R^2$, the result of adsorbing ammonium by natural zeolite was consistent with the presumption of Freundlich model.

The thermodynamic parameters related to the adsorption process, i.e., Gibb's free energy change ($\Delta G^0$, kJ·mol$^{-1}$), enthalpy change ($\Delta H^0$, kJ·mol$^{-1}$), and entropy change ($\Delta S^0$, J·mol$^{-1}$·K$^{-1}$) were also calculated respectively according to equations in Table S1, based on the adsorption equilibria at different temperatures, as shown in Table 5 and Figure 4c. Apparently, $\Delta G^0$ values were negative at all temperatures, indicating the NH$_4^+$ adsorption by natural zeolite to be a spontaneous process [42,43]. The $\Delta S^0$ values were also positive, which not only reflected that the randomness of the adsorption reaction increased with the elevating temperature, but also indicated that NH$_4^+$ maintained a high

affinity with natural zeolites [44]. The $\Delta H^0$ value was 21.36 kJ·mol$^{-1}$, revealing that the adsorption process was an endothermic reaction and the adsorption heat was relevant to the chemisorption process [42,43].

**Table 5.** The calculated thermodynamic parameters.

| T (K) | $K_c$ ($\times 10^{-3}$) | $\Delta G^0$ (kJ·mol$^{-1}$) | $\Delta H^0$ (kJ·mol$^{-1}$) | $\Delta S^0$ (J·mol$^{-1}$·K$^{-1}$) |
|---|---|---|---|---|
| 288 | 0.58 | −15.52 | 21.36 | 128.05 |
| 298 | 0.95 | −16.80 | | |
| 308 | 1.23 | −18.21 | | |
| 323 | 1.62 | −19.85 | | |
| 338 | 2.42 | −21.89 | | |

*3.5. Simulation of Zeolite's Ammonium Capture*

This work provided a thorough investigation of zeolite–ammonium interactions via MD simulation. The O, Si and Al atoms of the zeolite framework were all considered the possible adsorption sites for ammonium capture (Figure S2). The isomorphous substitution of aluminum atoms to Si ones resulted in insufficient positive charges within the crystal lattice, which were compensated by exchangeable cations in the intrinsic pore networks throughout the aluminosilicate crystallites. The existence of excessive net negative charges means some O atoms of the framework existed in the unbonded form, unlike the bridged Al-O-Si pattern. Therefore, both the bridged and unbonded O atoms were considered the independent sites and were performed with the corresponding calculations respectively.

This study also separately calculated the binding energies of the O, Si and Al sites in the framework of natural zeolite after adsorbing $NH_4^+$. It could be seen from the calculation results in Table 6 that the adsorption energy at each adsorption site was all negative, indicating that in theory $NH_4^+$ could be spontaneously adsorbed around the O, Si and Al atoms. In particular, the adsorption energy around the O atom was the faintest, indicating that $NH_4^+$ was adsorbed in the most stable way near this type of atom within the natural zeolites' skeleton. In the meantime, the extremely low value of adsorption energy also implied that the O-atom adsorption site presented chemisorption to $NH_4^+$ [31,32]. With such high level of binding energy, one could expect to remove ammonium difficultly from the structure during zeolite regeneration. As illustrated by the results of zeolite regeneration experiments (Figure 5b), according to different concentrations of regenerated saline solution, only 17~32% of ammonia release was observed, suggesting more ammonium was effectively trapped within the natural zeolites' pore structure by strong binding energies. Furthermore, as illustrated from the results of five adsorption-desorption cycles (Figure 5c), the removal ratio of $NH_4^+$ by natural zeolite decreased with the increase of adsorption-desorption cycles, due to the adsorption sites on zeolite were occupied and the natural Chinese zeolite was saturated constantly. However, the desorption ratio of adsorbed zeolite gradually increased with the increase of adsorption-desorption cycles in 2 mol/L NaCl solution, and even more than 95% of the desorption ratio was obtained in the fifth cycle. The phenomena were caused by the fact that the weakly-bond $NH_4^+$ was re-exchanged by Na$^+$ easily because the proportion of strong adsorption interactions such as chemical adsorption and hydrogen bonding decreased while that of relatively weak interactions such as electrostatic attraction and ion exchange increased with the occupation of $NH_4^+$ in the pores of zeolite continuously. Anyway, the cumulative adsorption capacity of $NH_4^+$ on zeolite after 5 cycles was great enough because the adsorbed $NH_4^+$ ions were not being desorbed completely in each desorption cycle, displaying its strong $NH_4^+$ adsorption capacity. Comprehensively considering, these results indicated the natural Chinese zeolite has great adsorption capability of high concentration $NH_4^+$, and has great potential to be used as the effective absorbing substance for fast removal and recovery of strength in the wastewater pretreatment unit of industrial coal chemical plant, finally to facilitate subsequent biochemical treatment to attain emission environmental standards.

**Table 6.** The adsorption energy of ammonium on the four possible adsorption sites in the zeolite.

| Adsorption Site | Bridged O Site | Unbonded O Site | Si Site | Al Site |
|---|---|---|---|---|
| $E_{\text{ads,cal}}$ (kJ/mol) | −86.77 | −136.77 | −40.90 | −61.32 |

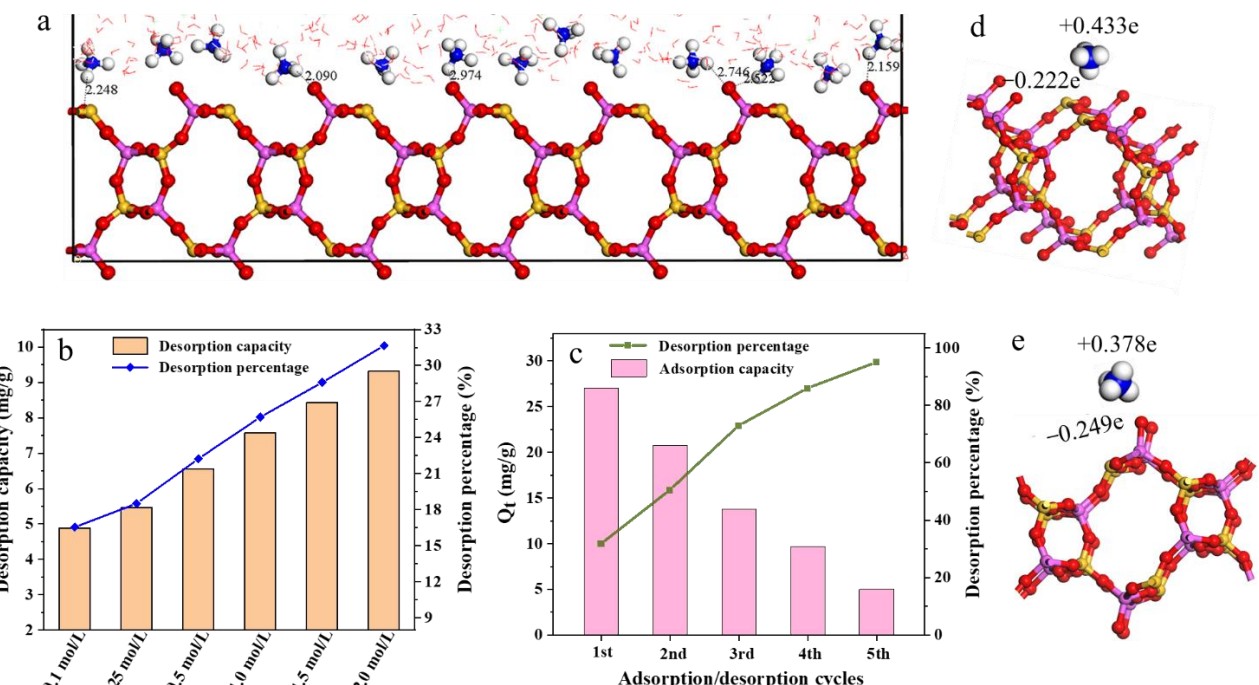

**Figure 5.** Visualization of ammonium adsorbed on the framework of natural zeolite. (**a**) Hydrogen bonding of ammonium ion with O atom in the zeolite framework; (**b**) Zeolite regeneration in NaCl solution with different concentrations; (**c**) Adsorption/desorption cycles using 2 mol/L of NaCl solution; (**d**) Hirshfeld charges between ammonium and the bridged O atom in the zeolite framework; (**e**) Hirshfeld charges between ammonium and the unbonded O atom in the zeolite framework. Yellow represents Si atoms, pink = Al, red = O, white = H, blue = N.

G. Gilli and P. Gilli both believed that the shortest hydrogen-bond length of all intermolecular N-H··O/O-H··N bonds was within the range of 2.532~2.660Å [45]. In this study, the calculated length of bond between the zeolite skeleton's O-atom and the adsorbed $NH_4^+$'s H-atom was 2.746~2.974Å (Figure 5a), indicating that there was a hydrogen bond between the $NH_4^+$ and the zeolite skeleton's O-atom, while the short bond-length of 2.090~2.522Å implied that the adsorption stronger than the hydrogen bond also existed between the adsorbed $NH_4^+$ and the zeolite skeleton's O-atom, which was highly likely to be chemisorption. Combined with the conclusions about the chemisorption in the previous Section 2.3, the calculation of the hydrogen bond's length not only theoretically verified the existence of a hydrogen bond between $NH_4^+$ and the surface of natural zeolite, but also facilitated to prove that a strong chemisorption also existed within the adsorption system.

The chemisorption of ammonium by natural zeolite represented the electron transfer between bonded ammonium's hydrogen atoms and the framework's oxygen atoms [38,39]. Hirshfeld charges of the adsorbed ammonium on the framework's bridged and unbonded oxygen sites were +0.433e and +0.378e respectively (Figure 5d,e), suggesting a stronger tendency of electron transfer from unbonded O atom to adsorbed ammonium. Therefore, these unbonded oxygen atoms presented the greatest preference for chemisorption of ammonium ions onto the zeolite surface.

### 3.6. Structural Changes of Natural Zeolite after Adsorption of Ultrahigh Concentration NH$_4^+$

To verify the effect on the structure of natural zeolite after adsorption of ultrahigh-concentration NH$_4^+$, the natural zeolite before and after adsorption of NH$_4^+$ was individually characterized by FTIR with the results shown in Figure 6. The peaks at 3000~4000 cm$^{-1}$ and ~1600 cm$^{-1}$ were the stretched and deformed vibrations of -OH adsorbed on the zeolite surface, respectively. The peak at 900~1100 cm$^{-1}$ belonged to the O-T-O bond vibration in the internal structure of zeolites, and the one at 750~790 cm$^{-1}$ was the symmetric stretching vibration of the T-O bond, where T represented Si and Al atoms. No stretching vibration of the N-H bond (3300~3500 cm$^{-1}$) was detected on the curve after NH$_4^+$ adsorption, indicating that a little NH$_4^+$ could hardly be detected. However, comparing the absorption peak positions of zeolites before and after NH$_4^+$ adsorption at 900~1100 cm$^{-1}$, it was found that the absorption peak of the O-T-O bond exhibited an obvious "blue shift". This indicated that the pressed became more stable after NH$_4^+$ adsorption by zeolites [46]. In the meantime, compared with the deformed vibration of -OH on the zeolite surface before and after NH$_4^+$ adsorption, a certain "red shift" occurred at ~1600 cm$^{-1}$, indicating that the adsorbed NH$_4^+$ would react with the -OH via hydrogen bonding to cause the -OH weakening its bond energy and moving to the low-frequency region. The above shifts of the absorption peaks not only demonstrated the actual formation of chemical bonds, but also verified the inference about the existence of chemical bonds in the previous experimental and theoretical results.

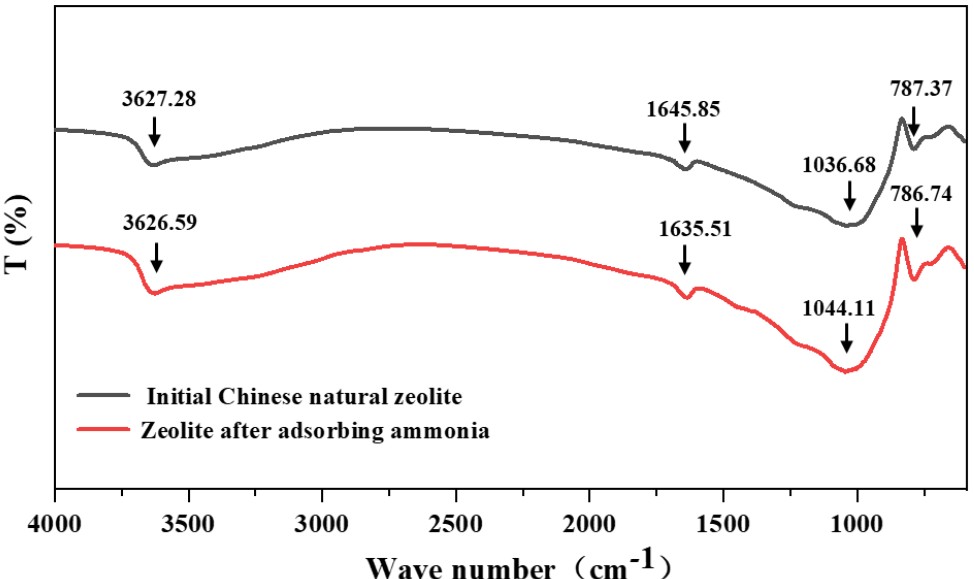

**Figure 6.** FTIR spectra of the Chinese natural zeolite before and after adsorption of high-concentrtion ammonia.

### 3.7. Elucidation of Mechanism in Ammonium Adsorption Process

By integrating adsorption kinetics, isotherms, thermodynamic analyses, and theoretical calculations, the mechanism of adsorption by natural zeolite to ultrahigh-concentration NH$_4^+$ was identified eventually. It was different from that of low-concentration NH$_4^+$ which was affected by a single ion-exchange process or electrostatic attraction. This mechanism was jointly affected by a variety of influencing factors, meaning that NH$_4^+$ in the adsorption system was mainly promoted by the synergy of the ion-exchange process, electrostatic attraction, and chemical adsorption. Meanwhile, hydrogen bonding also played an auxiliary role in the efficient adsorption by natural zeolite. Specifically, NH$_4^+$ in the zeolites' adsorption system ion completed the exchange process with the exchangeable metal cations (like Na$^+$, K$^+$, Ca$^{2+}$) in the zeolite structure before entering the pores of the zeolite. On the one hand, there was a stronger negative electric field in the zeolite structure, resulting in

a strong electrostatic attraction [35]. On the other hand, part of the adsorbed $NH_4^+$ was firmly bonded by the strong chemisorption at the O-atom adsorption site of the zeolite skeleton, which was not easy to be desorbed [28]. In addition, a considerable part of $NH_4^+$ adsorption would be subject to weaker hydrogen bonding. These adsorbed $NH_4^+$ were vulnerable to the external influence for breaking the bond, which displayed an obvious desorption phenomenon. Chemisorption and hydrogen bonding jointly enabled the adsorption capacity of natural zeolite to $NH_4^+$ to be higher than its IEC and exhibited a good adsorption ability. The adsorption mechanism of natural zeolite to ultrahigh-concentration $NH_4^+$ is shown in Figure 7.

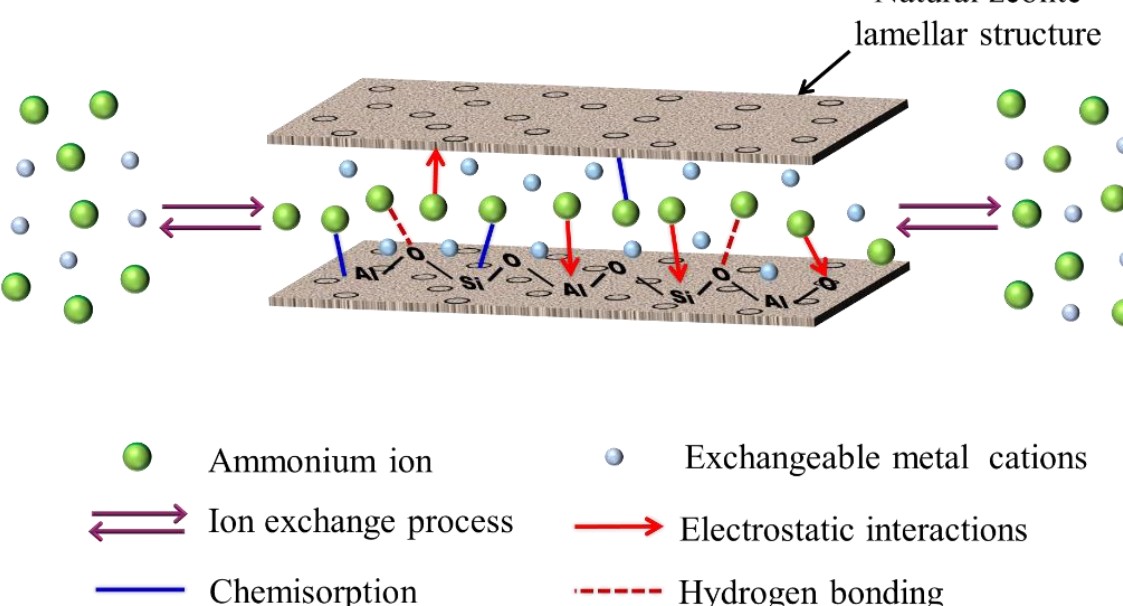

**Figure 7.** Schematic diagram for the adsorption mechanism of ammonium by the natural zeolite.

### 4. Conclusions

The adsorption behavior of natural zeolite to ultrahigh concentrations of $NH_4^+$ (1000~4000 mg-N/L) was significantly different from that of low-concentrations of NH4+ (<1000 mg-N/L). When the initial concentration was 4000 mg-N/L, the maximum adsorption capacity could reach 26.94 mg/g. Furthermore, the adsorption kinetics and isotherm analyses discovered that the adsorption process was consistent with both the pseudo-second-order kinetics model and the Freundlich model, indicating that the adsorptions to ultrahigh and high concentrations of $NH_4^+$ were carried out in the forms of monolayer and chemisorption on the surface of natural zeolite. The theoretical calculation results supported the existence of hydrogen bonds and chemisorption effect. The synergy of the ion exchange process, electrostatic attraction, and chemisorption played a leading role, while hydrogen bond exerted the auxiliary effect, which constituted the major adsorption mechanism by natural zeolite to ultrahigh-concentration $NH_4^+$. Overall, we have demonstrated that the Chinese natural zeolite possessed great potential for effectively capturing high-concentration ammonium in industrial wastewater, and this study has successfully enriched the research on ammonium's adsorption mechanism.

**Supplementary Materials:** The following supporting information can be downloaded at: https://www.mdpi.com/article/10.3390/w14152413/s1, Table S1: The representative equations of the studied kinetic and isotherm model and their parameters; Figure S1: The optimized structures of high-ammonium adsorption onto the natural zeolite; Figure S2: Visualization of ammonium ions adsorbed on different sites of the zeolite framework.

**Author Contributions:** Conceptualization, P.L., Y.L. and Z.L.; methodology, P.L., Y.L. and A.Z.; software, P.L. and A.Z.; validation, A.Z., Z.L. and X.L.; formal analysis, P.L., Z.L. and Y.L.; investigation, P.L. and Z.Y.; resources, L.Y. and Z.Y.; data curation, P.L. and X.L.; writing—original draft preparation, P.L., A.Z. and Y.L.; writing—review and editing, P.L., Z.L. and Y.L.; visualization, X.L. and L.Y.; supervision, Z.L.; project administration, Y.L., Z.L. and Z.Y. All authors have read and agreed to the published version of the manuscript.

**Funding:** This research was funded by National Natural Science Foundation of China, grant number 51978559 and the Key research and development project of Shaanxi province (Grant No. 2019ZDLSF05-06).

**Institutional Review Board Statement:** Not applicable.

**Informed Consent Statement:** Not applicable.

**Data Availability Statement:** Not applicable.

**Conflicts of Interest:** The authors declare no conflict of interest.

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
