# Peer review of "Adsorption Mechanism of High-Concentration Ammonium by Chinese Natural Zeolite with Experimental Optimization and Theoretical Computation"

_water, doi:10.3390/w14152413_

Round 1
Reviewer 1 Report
All my comments are in the attached file.

Reviewer 2 Report
Manuscript Number water-1814996-peer-review-v1
“Adsorption mechanism of high-concentration ammonium by Chinese natural zeolite with experimental optimization and theoretical computation”
The article is interesting and presents a series of carefully obtained data. The manuscript can be considered for publication after considering de following few recommendations:
- The abstract should be supported by a brief mention of the experimental conditions used; the range of NH4+concentration studied and the dosage of zeolite used… The value of the maximum adsorption capacity obtained must also be provided.
- Page 2 line 90-93: More details about the equations and models used to determine the textural characteristics (Pore volume, pore size-distribution…) of zeolite should be included.
- What is the volume of mesopores and that of micropores of Zeolite? Are mesopores more favorable for NH4+adsorption on zeolite or micropores? Add more discussion on the adsorption capacity of zeolite based on its composition.
- Please revise the title of Figure 1: a) SEM images and b) XRD patterns
- Please revise the title of Figure 5: (a) Hydrogen bonding of ammonium ion with O atom in the zeolite framework; (b) Zeolite regeneration in NaCl solution with different concentrations; (c) Hirshfeld charges between ammonium and the bridged O atom in the zeolite framework; (d) Hirshfeld charges between ammonium and the unbonded O atom in the zeolite framework.
- Page 5: What are the experimental conditions (initial NH4+ concentration, zeolite dosage) used in the experiments corresponding to Figures 1a and 1b. The discussion of the data in Figures 1a and 1b should be developed further.
- What do the solid lines and dotted lines represent in Figures 1a and 1b? Please specify
- Table 1: Under what conditions is the maximum adsorption capacity 29.48 obtained? The error in determining this value must be included. The authors report on page 5 line 172 and in conclusion that the maximum adsorption capacity reaches 27 mg/g when the initial NH4+ concentration is 4000 mg-N/L. Please specify

Reviewer 3 Report
1. The authors must provide novelty of the present work in the introduction section, as there are already published reports similar to the present study such as https://doi.org/10.1016/j.jhazmat.2009.09.156
Reviewer 4 Report
The manuscript entitled ''Adsorption mechanism of high-concentration ammonium by Chinese natural zeolite with experimental optimization and theoretical computation" includes a lot of work and contains many data. The experimental data are originally explained and interpreted. The work fits in your journal rigors and I recommend its publication.
I made the following recommendations:
- A comparison of the proposed zeolite with other reported zeolites on the basis of Langmuir maximum capacity of adsorption should be given.
- More considerations on the practical applicability of the targeted zeolite should be made.
- There some typing errors that should must be corrected.
